# Comparison of the Antibiotic Resistance of *Escherichia coli* Populations from Water and Biofilm in River Environments

**DOI:** 10.3390/pathogens13020171

**Published:** 2024-02-13

**Authors:** Aline Skof, Michael Koller, Rita Baumert, Jürgen Hautz, Fritz Treiber, Clemens Kittinger, Gernot Zarfel

**Affiliations:** 1Institute of Molecular Biosciences, University of Graz, 8010 Graz, Austria; aline.s@gmx.at (A.S.); fritz.treiber@uni-graz.at (F.T.); 2Diagnostic and Research Center for Molecular Biomedicine, Medical University of Graz, 8010 Graz, Austria; michael.koller@medunigraz.at (M.K.); rita.baumert@medunigraz.at (R.B.); juergen.hautz@gmail.com (J.H.); clemens.kittinger@medunigraz.at (C.K.)

**Keywords:** biofilm, ESBL, KPC-2, wastewater treatment plant, phenotyping

## Abstract

Antibiotic-resistant, facultative pathogenic bacteria are commonly found in surface water; however, the factors influencing the spread and stabilization of antibiotic resistance in this habitat, particularly the role of biofilms, are not fully understood. The extent to which bacterial populations in biofilms or sediments exacerbate the problem for specific antibiotic classes or more broadly remains unanswered. In this study, we investigated the differences between the bacterial populations found in the surface water and sediment/biofilm of the Mur River and the Drava River in Austria. Samples of *Escherichia coli* were collected from both the water and sediment at two locations per river: upstream and downstream of urban areas that included a sewage treatment plant. The isolates were subjected to antimicrobial susceptibility testing against 21 antibiotics belonging to seven distinct classes. Additionally, isolates exhibiting either extended-spectrum beta-lactamase (ESBL) or carbapenemase phenotypes were further analyzed for specific antimicrobial resistance genes. *E. coli* isolates collected from all locations exhibited resistance to at least one of the tested antibiotics; on average, isolates from the Mur and Drava rivers showed 25.85% and 23.66% resistance, respectively. The most prevalent resistance observed was to ampicillin, amoxicillin–clavulanic acid, tetracycline, and nalidixic acid. Surprisingly, there was a similar proportion of resistant bacteria observed in both open water and sediment samples. The difference in resistance levels between the samples collected upstream and downstream of the cities was minimal. Out of all 831 isolates examined, 13 were identified as carrying ESBL genes, with 1 of these isolates also containing the gene for the KPC-2 carbapenemase. There were no significant differences between the biofilm (sediment) and open water samples in the occurrence of antibiotic resistance. For the *E. coli* populations in the examined rivers, the different factors in water and the sediment do not appear to influence the stability of resistance. No significant differences in antimicrobial resistance were observed between the bacterial populations collected from the biofilm (sediment) and open-water samples in either river. The different factors in water and the sediment do not appear to influence the stability of resistance. The minimal differences observed upstream and downstream of the cities could indicate that the river population already exhibits generalized resistance.

## 1. Introduction

The presence of human-generated antibiotic resistance in surface waters is a fact of the present day. Almost every resistance mechanism known from the clinical environment has been detected in rivers, lakes, oceans, and even groundwater [1,2,3,4]. *Escherichia coli* (*E. coli*) is one of the most common facultative pathogens in human medicine and is also a key factor in the study of water quality, particularly in terms of fecal contamination. Studies have shown that a significant proportion of the *E. coli* population found in surface water already exhibits at least one form of acquired antibiotic resistance, and multi-resistant isolates are no longer a rarity [3,5,6,7,8,9,10].

Within surface waters, bacterial populations exist not only in the water itself but also on various surfaces within the water, such as sediment. The conditions for bacteria such as *E. coli* are different on these surfaces compared to open water and often result in the formation of a biofilm. Several characteristics of biofilms result in increased risk for the development of antibiotic resistance. In the biofilm, the transfer of antibiotic resistance-transmitting plasmids or phages is much easier because the cells physically live closer together. Therefore, antibiotic resistance genes are more likely to be spread in the population. Another factor is the presence of toxic substances that may accumulate in the sediment or on other surface structures (e.g., stones) and therefore become enriched in the biofilm as a result of sedimentation. A higher concentration of toxic substances can have a positive effect on the selection of antimicrobial resistance. Of note, this process does not occur only due to the presence of antibiotics or their degradation products. Other (toxic) substances, e.g., heavy metals or pesticides, can also contribute directly or indirectly to the stabilization of antibiotic resistance mechanisms [11,12,13,14].

However, some factors could lead to a low presence of antibiotic-resistant bacteria. In general, biofilms are more resistant to toxins and antibiotics, so there is less need for specific resistance mechanisms. In addition, competition with other (environmental) bacteria and the burden of the additional genetic load of antibiotic resistance genes are factors that counteract the stability of resistance. Bacteria in the biofilm are primarily from populations that have resided in the surface water for a long time and therefore have the time to lose ‘unnecessary’ resistance. In the water itself, the proportion of freshly introduced bacteria is higher. In the case of *E. coli*, for example, these come from humans and animals, including those that have just been or are being treated with antibiotics. Among them are highly resistant isolates as well [15,16,17].

Despite the potential impact of antibiotic-resistant bacterial development in surface waters, the factors that influence their development are not yet fully understood. This study sought to evaluate the impact of these effects on the presence of antibiotic-resistant bacteria in the *E. coli* population in biofilm/sediment as well as which specific resistance was present.

Two major rivers in southern Austria were chosen for sampling. Samples were taken from the river Mur, the main river in the state of Styria, where it flows through Graz. Samples from the Drava River, the main river in the state of Carinthia, were taken at its flow through Villach. For both rivers, downstream sampling took place past the wastewater treatment plant (WWTP). *E. coli* were isolated from the samples, and then, the isolates were tested for their susceptibility to 21 antibiotics. We then compared the populations of isolates from the water and biofilm with the general presence (proportion in the populations) of resistance and specifically to the individual antibiotics. In addition, the diversity of the isolated *E. coli* was investigated using a phenotypic differentiation system. On the one hand, this was used as a quality control to assess possible influences of the isolation of clones on the evaluation, and on the other hand, the aim was to offer a preliminary comparison of the composition between water and biofilm/sediment populations.

## 2. Materials and Methods

### 2.1. Sample Collection

Graz is the second largest city in Austria, with around 250,000 inhabitants; the Mur flows directly through the city center. Villach is the seventh largest city in Austria, with around 60,000 inhabitants, and is crossed by the Drau. The Mur and Drau do not flow through any major cities in Austria, and there is no city with more than 30,000 inhabitants before them.

The selection of sample locations was guided by the following criteria: the sites had to be situated at a minimum distance of 1.0 km from the WWTP, possess accessible sediment for sampling, and be feasible for examination. It was ensured that there were no heavy rain events in the area of the sample location three days before sampling.

Water samples were collected in 500 mL sterile plastic flasks (VWR International™, Vienna, Austria) 30 cm below the river surface and 50 cm from the bank. Sediment samples were lifted from the riverbed using a paddle and packed into sterile homogenizing bags (BagLight^®^ HD PolySilk^®^, Interscience, Saint Nom la Bretèche, France). It was ensured that mainly the superficial part of the sediment was harvested from a maximum of 5 cm deep. All samples were taken from the left bank of the river in the direction of flow.

On 24 November 2016, three water and three sediment samples were collected from the Mur River near the Weinzödlbrücke (W0X) upstream of Graz and the WWTP plant. The physico-chemical water parameters were as follows: water temperature 4.3 °C, pH-value 8.0, and oxygen content 11.2 mg/L. On 12 December 2016, three water and three sediment samples were collected from the Mur River in Kalsdorf (KD01) downstream of Graz and the WWTP. The physico-chemical water parameters were as follows: water temperature 3.7 °C, pH-value 7.7, and oxygen content 13.4 mg/L. On 3 April 2017, three water and three sediment samples were collected from the Drava River at Rennsteinerstraße (DR01) upstream of Villach and the WWTP there. The physico-chemical water parameters were as follows: water temperature 8.8 °C, pH-value 8.2, and oxygen content 11.3 mg/L. On 3 April 2017, three water and three sediment samples were collected from the Drava River at Klampfererweg (DK01) downstream of Villach and its WWTP (Table 1). The physico-chemical water parameters were as follows: water temperature 8.8 °C, pH-value 8.2, and oxygen content 11.3 mg/L.

### 2.2. E. coli Isolation from the Water Samples

The water samples were filtered using a pump (EZ-Stream™ Pump, Merck KGaA™, Darmstadt, Germany) and sterile membrane filters (EZ-Pak^®^ mixed cellulose ester filters, 47 mm, 0.45 µm, Millipore S.A.S 67120, Vienna, Austria). Approximately 100 mL from each sample was filtered into two portions of 50 mL each and placed onto Chromo Cult Coliform Agar plates (MERCK, Vienna, Austria). Additionally, 5 mL, divided into 500 µL portions of the water samples, was plated directly onto ten Chromo Cult Coliform Agar plates. This procedure was performed separately for each of the three samples from each sampling point. The Chromo Cult Coliform Agar plates were incubated at 42 °C for 24 h (h). All putative *E. coli* colonies (according to the manufacturer’s manual) were picked with sterile inoculating loops, transferred onto Columbia blood agar (BD™ BBL Stacker Plates, Heidelberg, Germany), and incubated for 24 h at 37 °C. Species identification for the isolates was performed through MALDI-TOF mass spectroscopy (Vitek^®^ MS, bioMérieux Austria™, Vienna, Austria). The confirmed *E. coli* were stored at −70 °C in bacterial storage flasks (mWE^®^ medical wire, Viabank, Corsham, UK).

### 2.3. E. coli Isolation from Sediment Samples

The sediment samples were diluted at 1:10 (1 g sediment + 9 mL Ringer Tween solution 0.3%) in 50 mL sterile plastic tubes (Greiner^®^, bio-One™, Kremsmünster, Austria) with 0.3% Ringer Tween (TWEEN 80^®^, Amresco, VWR™, Vienna, Austria) solution and incubated at the current river temperatures for one hour on a roll mixer (CATRM5^®^, servoLAB™, Kumberg, Austria). Approximately 500 µL of diluted samples was consequently plated onto Chromo Cult Coliform Agar plates. Post incubation, the colonies were isolated and stored using an identical procedure employed for the water samples.

### 2.4. Antimicrobial Susceptibility Testing

Agar diffusion tests were performed to determine antibiotic susceptibility for *E. coli* isolates as recommended by the European Committee on Antimicrobial Susceptibility Testing (EUCAST) for 21 antibiotics [18]. Tetracycline, chloramphenicol, and nalidixic acid susceptibility tests were carried out according to the Clinical Laboratory Standards Institute (CLSI) [19]. In brief, 2 mL of sterile sodium chloride solution (0.9%) were inoculated with one to three single colonies per isolate of fresh overnight cultures. The turbidity of these bacterial suspensions was adjusted to a turbidity of 0.5 ± 0.05 McFarland standard (DensiCheck, Biomérieux, Vienna, Austria). The suspension was spread with cotton swabs evenly over the surface of a Mueller Hinton (II) agar plate (bioMérieux, Vienna, Austria). Paper discs impregnated with differing antibiotics were consequently placed onto the agar surface. Following this, the agar plates were then incubated at 36 °C for 16 h. Interpretation of zone diameters was performed according to EUCAST or CLSI. To determine (clinical) resistance to colistin, protocols by Gales et al. and Boyen et al. were utilized [20,21].

Susceptibility testing was performed with ampicillin (10 µg), amoxicillin–clavulanic acid (20/10 µg), cefalexin (30 µg), cefuroxime (30 µg), cefoxitin (30 µg), cefotaxime (5 µg), piperacillin/tazobactam (30/6 µg), ceftazidime (10 µg), cefepime (30 µg), imipenem (10 µg), meropenem (10 µg), moxifloxacin (5 µg), ciprofloxacin (5 µg), nalidixic acid (30 µg), tetracycline (30 µg), tigecycline (15 µg), gentamicin (10 µg), amikacin (30 µg), trimethoprim/sulfamethoxazole (1.25/23.75 µg), colistin (10 µg), and chloramphenicol (30 µg) and BD BBL^TM^ Sensi-Disc^TM^ paper discs (BD, Vienna, Austria). *E. coli* ATCC 25922 served as the control strain in all analyses performed.

### 2.5. Phenotypic Confirmation of Extended-Spectrum Beta-Lactamases and Carbapenemases

To determine the minimum inhibitory concentrations (MIC) for imipenem and meropenem, Etest^®^ (bioMérieux Austria GmbH, Vienna, Austria) was used for all isolates that were resistant to at least one of the carbapenems tested. The expression of carbapenemases was confirmed through a modified Hodge Test.

Presumptive extended-spectrum beta-lactamases (ESBLs) were confirmed through double disc tests according to CLSI (30 μg ceftazidime, 30 μg cefepime, 30/10 μg ceftazidime–clavulanic acid, and 30/10 μg cefepime–clavulanic acid; bioMérieux Austria™, Vienna, Austria). KPC/MBL and OXA-48 Confirm Kit: Carbapenemases (Rosco Diagnostica, Taastrup, Denmark) determined the types of carbapenemases. Isolates that revealed an ESBL and/or carbapenemase phenotype were screened for antimicrobial resistance genes.

### 2.6. Determination of ESBL and Carbapenemase Genes

PCR detection and gene identification were performed for different β-lactamase gene families, *bla*_CTX-M-1-group_, *bla*_CTX-M-2-group_, *bla*_CTX-M-9-group_, *bla*_SHV_, and *bla*_TEM_. The PCR and sequencing procedures were performed as previously described and carried out for all isolates that showed an ESBL-positive phenotype [22,23,24,25]. For confirmation of *bla*_KPC_, PCR and sequencing protocols were used as previously described (Appendix A) [26].

In brief, DNA was extracted by boiling one colony suspended in 50 μL double-deionized water (95 °C for 10 min). After centrifugation for 1 min at 13,000 rpm (Centrifuge 5415 R, Eppendorf, Nijmegen, The Netherlands), the supernatant was used for the PCR reaction. Standard PCR protocols and conditions were modified in the following way: initial denaturation at 94 °C for 5 min; 35 cycles at 95 °C for 30 s, 52 °C for 45 s, and 72 °C for 60 s; and final incubation for 10 min at 72 °C using Taq DNA polymerase and dNTPs from QIAGEN (Hilden, Germany). Sequencing was performed with Eurofins overnight sequencing service (Eurofins, Hamburg, Germany).

### 2.7. Phenotyping—The PhenePlate System

To reveal the relationship between *E. coli* isolates, a biochemical fingerprint method (PhenePlate™, Stockholm, Sweden) was used according to the manufacturer’s protocol. The optical density at 620 nm (OD_620_) was measured after 16 h and 24 h using the Zenyth 3100 Multimode Detector^®^ platform (Anthos Mikrosysteme GmbH, Friesoythe, Germany). Analyses were performed using PhenePlate software Ver 7.1 (PhPlate AB, Stockholm, Sweden). Isolates with an identification level of 97.5% or higher were grouped into PHP clusters and considered as clones. To calculate the diversity level, the number of isolates in one sample set was divided by the number of clusters (singletons counted as one cluster with one member).

### 2.8. Statistics

Statistical analyses were conducted through IBM SPSS Statistics 27.0.1.0. To determine the *p*-values, Chi-squared tests (Fisher’s exact test) were performed. *p*-values less than 0.05 were considered statistically significant.

## 3. Results

In total, 831 *E. coli* were isolated from all samples from both rivers: 569 from the Mur River and 262 from the Drava River (Table 2, Appendix A). The isolates from the Mur consisted of 261 water and 308 sediment isolates. The Drava yielded 195 water and 67 sediment isolates (Table 2).

### 3.1. Comparison of Antibiotic Susceptibility of the Water and Sediment-Derived Isolates

The *E. coli* isolates from Mur River water showed a high proportion, 76.63% (200/261), with no resistance to any of the tested antibiotics. Resistance to one or two antibiotic classes was detected in 16.48% (43/261) of the isolates. Only 6.90% (18/261) of the isolates were multi-resistant, i.e., resistant to three or more classes of the antibiotics tested (Figure 1, Table 3).

Quite similarly, in the Mur sediment samples, 72.1% (222/308) of the isolates were susceptible to all tested antibiotics, 18.83% (58/308) were resistant, and 9.09% (28/308) were multi-resistant (Figure 1, Table 3).

The water isolates from the Drava revealed that 76.41% (149/195) were susceptible to all tested antibiotics, 17.95% (35/195) were resistant to one or two antibiotic, classes and 5.64% (11/195) were multi-resistant (Figure 1, Table 3).

The sediment isolates revealed that 76.12% (51/67) were susceptible to all tested antibiotics, 16.42% (11/67) were resistant to one or two antibiotic classes, and 7.46% (5/67) were multi-resistant (Figure 1, Table 3).

There were no significant differences in the occurrence of antibiotic resistance between the water and sediment samples in both the Mur and the Drava.

### 3.2. Comparison of Water Isolates from Upstream and Downstream of the WWTP

In the Mur water samples from upstream of the WWTP, 72.92% (105/144) of the isolates were susceptible to all antibiotics tested, 19.44% (28/144) were resistant to one or two antibiotic classes, and 7.64% (11/144) were multi-resistant. Of the isolates from downstream of the WWTP, 81.20% (95/117) showed no resistance to the tested antibiotics, 12.82% (15/117) showed resistance to one or two antibiotic classes, and 5.98% (7/117) were multi-resistant (Figure 2A; Table 3).

To conclude, for the Mur, there were no significant resistance differences between upstream and downstream of the WWTP.

The Drava water samples from upstream of the WWTP showed 83.33% (75/90) of the *E. coli* having no detected resistance, while 11.11% (10/90) of the isolates were resistant to one or two antibiotic classes. The multi-resistant isolates reached a proportion of 5.56% (5/90). Approximately 70.48% (74/105) of the water isolates from downstream of the WWTP showed no resistance to the tested antibiotics. In comparison, 23.81% (25/105) of the isolates were resistant to one or two antibiotic classes and 5.71% (6/105) were multi-resistant (Figure 2B; Table 3).

Thus, the Drava water isolates from upstream of the WWTP showed a significantly higher proportion of isolates with no resistance compared to downstream (83.33% vs. 70.48%, *p*-value = 0.042). In line with this, the downstream samples showed a significantly higher proportion of resistant isolates than the upstream samples (23.81% vs. 11.11%, *p*-value = 0.025).

### 3.3. Comparison of Sediment Isolates from Upstream and Downstream of the WWTP

64.60% (73/113) of the isolates from the Mur sediment samples from upstream of the WWTP showed no resistance to any of the tested antibiotics, 30.09% (34/113) were resistant to one or two antibiotic classes, and 5.31% (6/113) were multi-resistant. In the sediment samples from downstream of the WWTP, 76.41% (149/195) of the isolates were susceptible to all antibiotics tested, 12.31% (24/195) were resistant to one or two antibiotic classes, and 11.28% (22/195) were multi-resistant (Figure 2A; Table 3).

The proportion of isolates with no resistance to any tested antibiotics was significantly higher (*p*-value = 0.035) in the downstream samples compared to the upstream samples. In agreement with this, the proportion of isolates resistant to one or two antibiotic classes was significantly higher (*p*-value < 0.001) in the upstream samples. However, the sediment samples from downstream had a higher proportion of multi-resistant isolates than the upstream samples, although this difference in multi-resistance was not significant (*p*-value = 0.1).

In the Drava, there were no significant differences between the up- and downstream sediment samples.

The isolates of sediment samples from upstream of the WWTP showed 72.73% (24/33) having no detected resistance, while 15.15% (5/33) were resistant to one or two antibiotic classes, and 12.12% (4/33) were multi-resistant. The sediment samples from downstream of the WWTP showed 79.41% (27/34) isolates with no detected resistance, while 17.65% (6/37) were resistant, and 2.94% (1/37) were multi-resistant (Figure 2B; Table 3). None of the differences in the resistance patterns were significant.

### 3.4. Resistance to the 21 Antibiotics

The most common resistance in *E. coli* water and sediment isolates in both rivers was resistance to the β-lactam antibiotic ampicillin. This resistance was almost equally detected in all populations (Figure 3, Appendix A).

The greatest differences in the proportion of resistance were found between the Mur water and sediment isolates regarding the resistance to nalidixic acid. Approximately 10.34% (27/261) of the Mur water isolates were resistant to nalidixic acid compared to 15.91% (49/308) of the sediment isolates. In the Drava, a high proportion of isolates were also resistant to nalidixic acid: 9.23% (18/195) of the water isolates were resistant to the antibiotic, in contrast to only 5.97% (4/67) of the sediment isolate samples (Figure 3, Appendix A).

Resistance to aminoglycosides (amikacin and gentamycin) was rare in both rivers, but one isolate from the Drava showed resistance to the last line antibiotic amikacin. All isolates were susceptible to colistin, tigecycline, and carbapenems, with one exception: one isolate from the Mur sediment showed resistance to meropenem and imipenem (Appendix A).

The only significant difference between the water and sediment samples on the level of resistance to single antibiotics was the resistance to gentamicin for isolates from the Mur water and sediment samples (*p*-value = 0.043; Figure 3A, Appendix A).

### 3.5. Phenotyping of E. coli from the Mur and Drava Rivers

Phenotypic differentiation of all isolates through evaluation of metabolic reactions was performed using the PhenePlate (PhP) system.

For the Mur River, differentiation resulted in 87 PhP clusters consisting of 75.92% (432/569) of all isolates and 24.08% (137/569) singletons (Appendix A).

Over half of the clusters, 55 of the 87, occurred in more than one sample. Fifteen of these clusters had only members from upstream water and sediment, and 23 clusters consisted exclusively of downstream isolates. Three clusters corresponded to water isolates only (up- and downstream), and one consisted of sediment isolates only. There were 11 clusters with members from three samples, and two clusters (M-14 and M-18) had members from all four Mur samples (Appendix A).

Cluster M-18 was the largest Mur cluster with 27 isolates. In total, nine clusters consisted of ten or more isolates, and all of them had members of at least two different samples, always with a water sample and a sediment sample (Appendix A).

The Drava River isolate differentiation resulted in 33 clusters, including 75.19% (197/262) of all isolates and 65 (24.8%) singletons.

Approximately 29 of the 33 clusters occurred in more than one sample. Five of these clusters had only members from upstream water and sediment, and another five clusters had only members from downstream of the WWTP. Approximately seven clusters consisted of water isolates only (up- and downstream), 11 clusters had members of three samples, and one cluster (D-01) had members from all Drava samples (Appendix A).

With a total of seven isolates, cluster D-01 was the smallest by far. In total, 6 clusters consisted of at least 10 isolates. Only one of these large clusters, D-27, consisted exclusively of isolates from the downstream sediment, while the other five clusters included isolates from more than one sample. The largest cluster was D-30 with 45 isolates in total, all from upstream. Therefore, the influence of clusters D-27 and D-30 must be given special consideration in further analyses of diversity (Appendix A).

The overall diversity of the Mur River isolates was 2.33 isolates per cluster and 2.7 isolates per cluster for the Drava River.

The diversity between the water samples was lower (i.e., more isolates per cluster) at 2.05 (Mur) and 2.50 (Drava) than in the sediment samples, which were 1.75 (Mur) and 1.67 (Drava). Also, in both rivers, the upstream population had lower diversity than the downstream one. In this comparison, the highest difference was detected between upstream Mur isolates with 2.82 and downstream isolates with only 1.75. The isolates from the Drava River revealed 2.51 isolates per cluster upstream and 2.09 downstream (Table 4 and Table 5).

Perhaps due to the relatively small number of Drava sediment isolates, including a large cluster of 12 isolates (D-27), this sample set does not follow the general trend, and these values should therefore be taken with caution (Table 5).

### 3.6. ESBL and Carbapenemase-Producing E. coli

In total, 13 *E. coli* isolates with a presumptive ESBL phenotype according to susceptibility testing were confirmed via the CLSI-test as ESBLs. These ESBL isolates were present in all four sampling locations in the water samples but could only be isolated from sediment samples from the Mur River. In addition, one of these *E. coli* isolates (KD01EC110) revealed resistance to imipenem and meropenem. After a Rosco test, it was considered most likely to be a KPC producer (Table 6).

The resistance genes of isolates showing these phenotypes were then analyzed through sequencing. Among the detected ESBL genes, eight corresponded to *bla*_CTX-M-15_, three corresponded to *bla*_CTX-M-1_, one corresponded to *bla*_CTX-M-14,_ and one corresponded to *bla*_SHV-12_. Genes for the non-ESBL ß-lactamase TEM-1 were detected in six *E. coli* isolates.

The carbapenemase *bla*_KPC-2_ was genetically confirmed in the isolate KD01EC110 in combination with *bla*_CTX-M-1_ and *bla*_TEM-1_.

## 4. Discussion

Many studies show that antibiotic resistance can be found in nearly every type of environment [27]. Rivers and lakes seem to play a key role, as the wastewater from cities and hospitals eventually flows into public waters. Previous research has suggested that river sediments contribute to the prolonged persistence of various bacteria and resistance mechanisms, more so than in open river waters [17,28]. In the current study, we demonstrate that three-quarters of the tested *E. coli* Mur River (water and sediment) isolates were susceptible to all tested antibiotics. When compared to studies conducted in other locations, the proportion of resistant and multi-resistant *E. coli* isolates in the Drava River and the Mur River is low [5,6,29,30,31,32,33]. This could be related to the fact that the Mur and Drava flow through only a few major cities, and that waste-water treatment processes in Austria maintain a very high standard of quality.

Somewhat surprisingly, the findings indicate that the presence of resistant and especially multi-drug resistant *E. coli* in the area downstream of the WWTP did not differ from the sections taken from upstream of the river near a treatment plant. Only the water from the Drava River had a significantly higher proportion of *E. coli* resistant to one or two antibiotic classes, possibly due to the additional city flow through Villach and its WWTP. However, no resistance to a specific antibiotic was significantly increased. Such a change would be more likely for the Mur. Both the Drau and the Mur have roughly the same flow rate at their respective reaches (approx. 90 m^3^/s). However, Graz is a much larger city, with 250,000 inhabitants, than Villach, with approx. 60,000 inhabitants. Furthermore, the wastewater treatment plant can also play a decisive role here, with resistant *E. coli* strains being reduced or enriched to varying degrees in different wastewater treatment plants compared to non-resistant *E. coli*. This also applies to the introduction of other substances that can influence the occurrence of antibiotic-resistance in the water and in the biofilm. Other studies have already shown this [6,33].

Our sampling strategy intentionally avoided collecting samples from the wastewater stream and instead intended to show the general and long-term effects of upstream WWTP discharge. These results align with other studies showing that the impact of wastewater treatment plants tends to be significant, typically on rare resistance mechanisms (like carbapenemases), only in the area of the direct wastewater plume—that is, unless the river is small and/or largely untouched by human influence before wastewater inflow [5,6,7,34,35,36].

However, no significant differences in the resistance patterns in the sediment and river water samples were found. This could be due to the contamination of the sediment samples by river water, which unfortunately cannot be avoided during sampling. Also, many of the *E. coli* populations in the open water may originate from sediment and biofilm and only a small proportion of them represent freshly introduced *E. coli* (as evidenced by the comparison before and past the treatment plants). These two factors may be reasons why the sediment samples showed slightly higher variability in phenotypic clustering as well. Overall, the phenotypic relationship analysis shows, with one exception, that there is no strong clustering of a clone in either sample material. This high variability reinforces the conclusions drawn from the resistance data.

One noteworthy observation, not anticipated in our study planning but arising from the unique conditions of the course of the two rivers, was that the *E. coli* sediment samples before and after Villach did not differ significantly in their antibiotic resistance patterns. In the Mur, however, the isolates from the sediment samples downstream of Graz and its WWTP were significantly more susceptible to the antibiotics tested. This sediment sample from the Mur River was the only one that was collected from an area without a power station in the immediate area, thereby remaining unaffected by its flow velocity. The sinking—or rather the lack of sinking—of particles from the river due to damming (and thus reduced introduction of toxic substances) could be the cause of this finding. However, due to the design of the study and the limited number of sampling points, this is purely speculative.

In this study, ESBL- and KPC-producing *E. coli* isolates were detected without the use of selective media with added antibiotics in culture. The KPC-2-producing isolate found was the first *E. coli* with these resistance mechanisms to be isolated from the environment in Austria and the first KPC-producing isolate ever found in a river in Austria [37]. Previous studies of river water and sediment isolates always used selective media to detect these kinds of resistance genes in different bacteria. Furthermore, in parallel isolation without antibiotics, the same studies failed to isolate carbapenem-resistant *E. coli* [5,38,39,40,41]. This is particularly interesting, as clinical studies in Austria also show that carbapenemase-producing *E. coli* are extremely rare. However, it is possible that KPC-forming Enterobacteriaceae are better able to persist in sediment (or in the biofilm found there) than was previously suspected or that the colonization of the normal healthy population is much higher than the clinical data suggest [42,43].

## 5. Conclusions

This study found no significant impact of the sample location (water or sediment) on antibiotic resistance. Thus, at least in the case of the *E. coli* population, the differences between open water and sediment do not seem to have a significant impact on the stability of antibiotic resistance. Of course, this cannot be generalized. The situation may be different in rivers with varying sewage loads or countries with different resistance situations.

The presence of a KPC-2-producing isolate shows that even resistance to last-line antibiotics or multi-resistant isolates can persist, at least temporarily, in the environment. Hence, the possibility of colonization or even infection with such challenging-to-treat strains through the use of contaminated water is now a concern. In particular, these strains might establish themselves permanently in surface waters, posing an ongoing risk even after measures are implemented to prevent the further introduction of such strains.

These findings offer valuable insights into the occurrence of antibiotic resistance within the environment, underscoring the potential threat of their enduring integration as permanent denizens of aquatic ecosystems. The data enrich our understanding of how antibiotic resistance establishes itself in river systems and underscore the imperative need for ongoing vigilance and preventive strategies to effectively tackle this concern.

## Figures and Tables

**Figure 1 pathogens-13-00171-f001:**
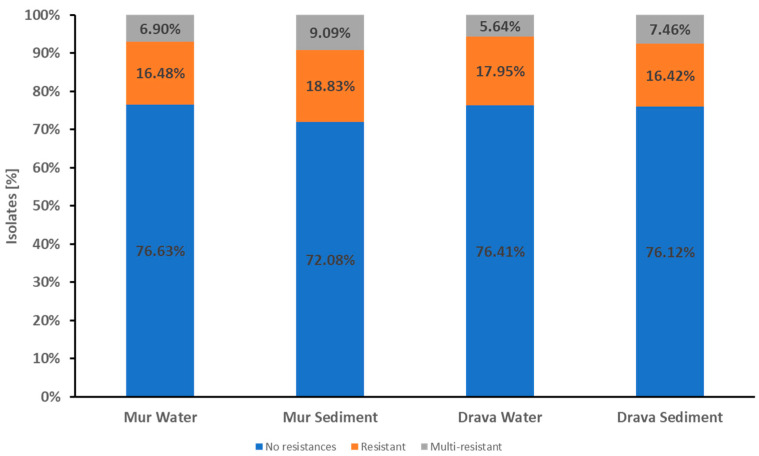
Proportions of resistance and multi-resistance in *E. coli* isolates from Mur and Drava water and sediment samples. The stacked columns represent the proportions of isolates showing the respective phenotype in all water and sediment samples. Blue columns indicate isolates susceptible to all antibiotics tested. Isolates with resistance to one or two classes of the tested antibiotics were classified as resistant (indicated in the orange part of the columns). Resistance to three or more classes of the tested antibiotics was classified as multi-resistance (indicated in the gray part of the columns).

**Figure 2 pathogens-13-00171-f002:**
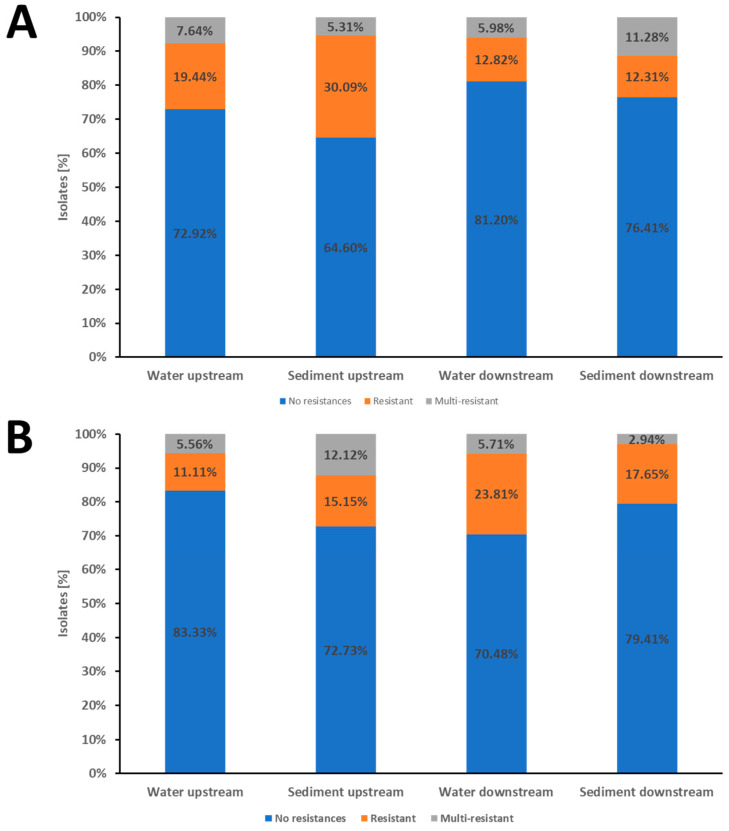
Proportions of resistance and multi-resistance in *E. coli* isolates. Panel (**A**): Isolates from the Mur River. Panel (**B**): Isolates from the Drava River. Panel (**A**,**B**): The stacked columns represent the proportions of isolates showing the respective phenotype in the water and sediment samples from upstream and downstream of Graz (**A**) or Villach (**B**) and its WWTP. Blue columns indicate isolates susceptible to all antibiotics tested. Isolates with resistance to one or two classes of the tested antibiotics were classified as resistant (indicated in the orange part of the columns). Resistance to three or more classes of the tested antibiotics was classified as multi-resistant (indicated in the gray part of columns).

**Figure 3 pathogens-13-00171-f003:**
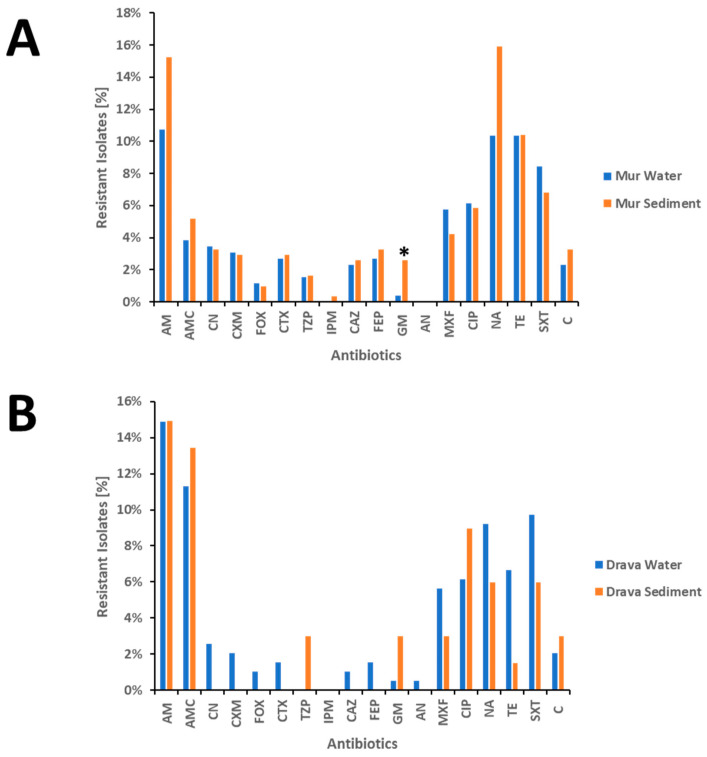
Proportions of *E. coli* isolates resistant to the tested antibiotics. Panel (**A**): Isolates from the Mur River. Panel (**B**): Isolates from the Drava River. Panel (**A**,**B**): Blue columns indicate isolates from the water samples, and orange columns indicate isolates from the sediment samples. * indicates a significant difference with a *p*-value less than 0.05. AM, ampicillin; AMC, amoxicillin–clavulanic acid; TZP, piperacillin/tazobactam; CN, cephalexin; CXM, cefuroxime; FOX, cefoxitin; CTX, cefotaxime; CAZ, ceftazidime; FEP, cefepime; AN: acrylonitrile; IPM, imipenem; CIP, ciprofloxacin; MXF, moxifloxacin; GM, gentamicin; SXT, trimethoprim/sulfamethoxazole; TE, tetracycline; NA, nalidixic acid; C, chloramphenicol.

**Table 1 pathogens-13-00171-t001:** List of sampling sites including the sampling date, site name, geographic name, and coordinates.

Sampling Date	Sample Name	Location	Coordinates
24 November 2016	W0X	Graz, Weinzödlbrücke	47°06′30.3″ N 15°23′25.4″ E
21 December 2016	KD01	Kalsdorf	46°58′01.7″ N 15°29′24.6″ E
3 April 2017	DR01	Villach, Rennsteinerstraße	46°38′30.7″ N 13°48′21.1″ E
3 April 2017	DK01	Villach, Klampfererweg	46°36′46.0″ N 13°55′16.3″ E

**Table 2 pathogens-13-00171-t002:** Overview of the numbers of *E. coli* isolates isolated from all samples from the rivers Mur and Drava.

	Mur			Drava		
	Water	Sediment	Sum of Isolates	Water	Sediment	Sum of Isolates
**Upstream**	144	113	257	90	33	123
**Downstream**	117	195	312	105	34	139
**Sum of isolates**	261	308	569	195	67	262

**Table 3 pathogens-13-00171-t003:** Proportions of resistance and multi-resistance in *E. coli* isolates from Mur and Drava water and sediment samples. A *p*-value less than 0.05 was considered statistically significant.

	Mur Water	Mur Sediment	*p*-Value	Drava Water	Drava Sediment	*p*-Value
	(261 isolates)	(308 isolates)		(195 isolates)	(67 isolates)	
No resistance	76.63% (200)	72.08% (222)	0.25	76.41% (149)	76.12% (51)	1
Resistant	16.48% (43)	18.83% (58)	0.51	17.95% (35)	16.42% (11)	0.85
Multi-resistant	6.9% (18)	9.09% (28)	0.36	5.64% (11)	7.46% (5)	0.56
	**Mur water us**	**Mur water ds**	***p*-value**	**Drava Water us**	**Drava water ds**	***p*-value**
	(144 isolates)	(117 isolates)		(90 isolates)	(105 isolates)	
No resistance	72.92% (105)	81.2% (95)	0.14	83.33% (75)	70.48% (74)	0.04
Resistant	19.44% (28)	12.82% (15)	0.18	11.11% (10)	23.81% (25)	0.02
Multi-resistant	7.64% (11)	5.98% (7)	0.63	5.56% (5)	5.71% (6)	1
	**Mur sediment us**	**Mur sediment ds**	***p*-value**	**Drava sediment us**	**Drava sediment ds**	***p*-value**
	(113 isolates)	(195 isolates)		(33 isolates)	(34 isolates)	
No resistance	64.6% (73)	76.41% (149)	0.03	72.73% (24)	79.41% (27)	0.58
Resistant	30.09% (34)	12.31% (24)	<0.01	15.15% (5)	17.65% (6)	1
Multi-resistant	5.31% (6)	11.28% (22)	0.1	12.12% (4)	2.94% (1)	0.2
	**Mur water us**	**Mur sediment us**	***p*-value**	**Drava water us**	**Drava sediment us**	***p*-value**
	(144 isolates)	(113 isolates)		(90 isolates)	(33 isolates)	
No resistance	72.92% (105)	64.6% (73)	0.17	83.33% (75)	72.73% (24)	0.21
Resistant	19.44% (28)	30.09% (34)	0.06	11.11% (10)	15.15% (5)	0.54
Multi-resistant	7.64% (11)	5.31% (6)	0.61	5.56% (5)	12.12% (4)	0.25
	**Mur water ds**	**Mur sediment ds**	***p*-value**	**Drava water ds**	**Drava sediment ds**	***p*-value**
	(117 isolates)	(195 isolates)		(105 isolates)	(34 isolates)	
No resistance	81.2% (95)	76.41% (149)	0.4	70.48% (74)	79.41% (27)	0.38
Resistant	12.82% (15)	12.31% (24)	1	23.81% (25)	17.65% (6)	0.64
Multi-resistant	5.98% (7)	11.28% (22)	0.16	5.71% (6)	2.94% (1)	1

Abbreviations: us—upstream and ds—downstream of the cities and their WWTP.

**Table 4 pathogens-13-00171-t004:** The average number of isolates per cluster (including singletons) for the sample set from the Mur River.

Mur River	Water Isolates	Sediment Isolates	Total Isolates
Upstream	2.25	2.22	2.82
Downstream	1.60	1.37	1.75
Total course	2.06	1.75	2.33

**Table 5 pathogens-13-00171-t005:** Average number of isolates per cluster (including singletons) for the sample set from the Drava River.

Drava River	Water Isolates	Sediment Isolates	Total Isolates
Upstream	2.50	1.27	2.51
Downstream	1.81	1.95	2.09
Total course	2.50	1.67	2.70

**Table 6 pathogens-13-00171-t006:** Detected resistance genes, resistance patterns, and the PHP cluster (or singletons) in which the isolates occur with ESBL- and KPC-harboring *E. coli* isolates from water and sediment.

Isolate ID	Origin	PHP Cluster	Resistance Genes	Resistance Pattern ^1^
DK01EC050	water	M-20	*bla* _CTX-M-1_	AM, CN, CXM, CTX, FEP
DR01EC012	water	M-31	*bla*_SHV-12_, *bla*_TEM-1_ ^2^	AM, AMC, CN, CXM, FOX, CAZ, GM, MXF, CIP, NA, SXT, TE, C
DR01EC036	water	M-31	*bla* _CTX-M-15_	AM, AMC, CN, CXM, CTX, CAZ, FEP, TE
KD01EC006	sediment	Single.	*bla* _CTX-M-14_	AM, AMC, CN, CXM, CTX, FEP, MXF, CIP, NA, SXT
KD01EC110	sediment	Single.	*bla*_CTX-M-1_, *bla*_TEM-1_*, bla*_KPC-2_	AM, AMC, CN, CXM, FOX, CTX, TZP, CAZ, FEP, MEM, IPM, MXF, CIP, NA, C
KD01EC112	sediment	M-42	*bla* _CTX-M-15_ *, bla* _TEM-*1*_	AM, AMC, CN, CXM, CTX, CAZ, FEP, MXF, CIP, NA
W04EC016	sediment	M-18	*bla* _CTX-M-15_	AM, CN, CXM, CTX, CAZ, FEP, NA, SXT, TE
W04EC018	sediment	M-18	*bla*_CTX-M-15_, *bla*_TEM-1_	AM, AMC, CN, CXM, CTX, CAZ, FEP, NA, SXT, TE
W04EC029	sediment	M-18	*bla* _CTX-M-15_	AM, CN, FOX, CTX, CAZ, FEP, NA, SXT, TE
W04EC057	sediment	Single.	*bla* _CTX-M-1_	AM, CN, CXM, CTX, CAZ, FB, TE
W04EC088	water	M-18	*bla*_CTX-M-15_, *bla*_TEM-1_	AM, CN, CXM, CTX, CAZ, FEP, NA, SXT, TE
W04EC090	water	M-18	*bla* _CTX-M-15_	AM, CN, CXM, CTX, CAZ, FEP, NA, SXT, TE
W04EC093	water	M-18	*bla* _CTX-M-15_	AM, CN, CXM, CTX, CAZ, FEP, NA, SXT, TE

^1^ AM, ampicillin; AMC, amoxicillin–clavulanic acid; TZP, piperacillin/tazobactam; CN, cephalexin; CXM, cefuroxime; FOX, cefoxitin; CTX, cefotaxime; CAZ, ceftazidime; FEP, cefepime; MEM, meropenem; IPM, imipenem; CIP, ciprofloxacin; MXF, moxifloxacin; GM, gentamicin; SXT, trimethoprim/sulfamethoxazole; TE, tetracycline; NA, nalidixic acid; C, chloramphenicol; ^2^ Resistance gene *bla*_TEM-1_ encoding non-extended-spectrum β-lactamases.

## Data Availability

Data are available within the article.

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
