# Peer review of "Comparison of the Antibiotic Resistance of Escherichia coli Populations from Water and Biofilm in River Environments"

_pathogens, 2024, doi:10.3390/pathogens13020171_

Round 1
Reviewer 1 Report
Comments and Suggestions for Authors
The manuscript entitled “Comparison of antibiotic resistance of Escherichia coli populations from water or sediment in river environments” have demonstrated significant results including estimation of antibiotic resistance of E. coli population.
Authors should correct manuscript according to the suggestion.
Minor issues:
line 122 please specify what confirmatory tests were performed to confirm the identification of E. coli
lines 137 and 139 why are EUCAST and CLSI standards used to determine antimicrobial properties?
Line 353: It should be citation for this statement
References
Ref. 7, 10-15, 25, 29, 31, 33, 35-37: journals name should be given as abbreviation
Author Response
Reviewer 1
Answer: Thanks for the comments and the help to improve our manuscript. We have made the improvements suggested by the reviewer.
Minor issues:
R1: line 122 please specify what confirmatory tests were performed to confirm the identification of E. coli
Answer: Thanks for the comment. the sentence “Species identification for isolates was performed through MALDI-TOF mass spectroscopy” has been moved to the paragraph to make the correct flow of the experiments clearer.
R1: lines 137 and 139 why are EUCAST and CLSI standards used to determine antimicrobial properties?
Answer: In order to ensure better comparability with clinical isolates in Austria and Europe, we have focused on the commonly used standard (EUCAST), but methods according to EUCAST are not available for all antibiotics tested. Therefore, CLSI was used in these cases.
R1: Line 353: It should be citation for this statement
Answer: Citations were added.
References
Ref. 7, 10-15, 25, 29, 31, 33, 35-37: journals name should be given as abbreviation
Answer: were changed
Reviewer 2 Report
Comments and Suggestions for Authors
The purpose of this work was to examine resistance of facultative pathogen bacterial species Escherichia coli on various antibiotic types. The bacteria had been recovered from fresh water and sediments in cities of Gratz and Villach, Mura and Drava rivers specifically.
The samples had been collected upstream and downstream, away from any populated areas. Downstream sampling was carried out 1Km after the last waste water purifier. Samples of water columns, sediments, as well as biofilm scraps from various rock surfaces were recovered on chosen locations as the authors wanted to examine is there any difference in E. coli strains and their resistance level, considering the matrix they been isolated from.
Further, they isolated E. coli types from previously collected samples using membrane filtration as well as selective chromogenic surface cultivation. The same was used with biofilm and sediment samples after specific preparation of these samples.
Identifications of E. coli were conducted with MALDI TOF method.
Standardized tests like EUCAST and CLSI had been used for antimicrobial sensitivity ratios. Phenotypic determination with dedicated sets. Genetic identification with PCR technique, as well as referred works protocols.
It has been 831 strains of E. coli isolated in these tests. All were tested for resistance to 21 antibiotics from seven different classes.
This study did not establish a significant influence of the matrix on bacterial resistance. The research showed that there is no statistically significant difference between the amount and type of resistant strains of E. coli among the matrices (water, sediment) as well as between examined locations.
Results had shown that isolated strains of E. coli are resistant on at least one of mentioned antibiotics, but as called last line multi resistant strains of E. coli had been detected.
The results showed that the isolated E. coli strains were resistant to at least one of the tested antibiotics. However, multi resistant strains and strains from the so-called last line were also detected.
General comments
The authors believe that the obtained data enriches the understanding of how antibiotic resistance is established in river systems and helps in thinking about a preventive strategy for solving the problem of eventual permanent integration of multi resistant strains in river environments.
Authors were concerned about the presence of these multi resistance strains of E. coli in environment and possibility of infections and colonisations in human and animal populations, caused by the use of contaminated river water.
It is possible to establish monitoring of bacterial resistance throughout the seasons as addition to monitoring of physical and chemical parameters of a water (such as pH, temperature, presence of nutrients, etc.), which would, maybe, record statistically significant differences in the presence of resistant E. coli between matrices/seasons/ locations.
I refer on the research below to point the authors on their work in future:
Monitoring of Water Quality, Antibiotic Residues and Antibiotic-Resistant Escherichia coli in the Kshipra River in India over a 3-Year Period; Nada Hanna, Manju Purohit, Vishal Diwan, Salesh P. Chandran, Emilia Riggi, Vivek Parashar, Ashok J. Tamhankar and Cecilia Stålsby Lundborg; International Journal of Environmental Research and Public Health; Published: 22 October 2020.
Special comments:
Line 18
Please, explain why you didn't mention the biofilm in the title?
Line 31
Please, explain do you consider sediment and biofilm to be the same matrix?
Line 38/39
Please, delete the repeated sentence from Line 34/35.
Line 100
Please, explain do you consider that samples which are scraped from a stone as biofilm? Please, reconsider for another term "Periphyton".
"Periphyton or fouling represents an integral and independent microecosystem in aquatic ecosystems, which carries biotic components such as algae, fungi, bacteria, protozoa, metazoa together with abiotic components such as substrate, extracellular polysaccharides and detritus."
Line 116
Please, explain abbreviation COL?
Line 122
Did you confirm E. coli after they grew on blood agar? Which test - MALDI TOF?
Line 125-129
Please, explain at what depth did you grab the sediment? How the samples from the stones were collected and on what surface?
According to which protocol or publication did you process samples of sediment and stones? Please provide the reference.
Line 140
Please, explain how did you prepare the E. coli suspension for Muller Hinton agar? What turbidity did you use?
Line 165-169
Please, explain which PCR detection and gene identification you performed?
Line 172
Please, write the full name of the abbreviation "OD620".
Line 185
Please, reconsider whether Table 1 should be referred here, given that you are listing data from Table 2.
Table 2
Please, explain why in this table there is no separate data for E. coli strains isolated from biofilm as you stated.
Line 190-204
The results described in this paragraphs do not correspond to the data in Figure 1 (Panels A and B). Consider making separate figure for the described results.
Line 216-235
Please, reconsider to create a table with the results from the text in order to highlight statistically significant differences which you should then explain in the discussion.
Author Response
Reviewer 2
General comments
R2: The authors believe that the obtained data enriches the understanding of how antibiotic resistance is established in river systems and helps in thinking about a preventive strategy for solving the problem of eventual permanent integration of multi resistant strains in river environments.
Authors were concerned about the presence of these multi resistance strains of E. coli in environment and possibility of infections and colonisations in human and animal populations, caused by the use of contaminated river water.
It is possible to establish monitoring of bacterial resistance throughout the seasons as addition to monitoring of physical and chemical parameters of a water (such as pH, temperature, presence of nutrients, etc.), which would, maybe, record statistically significant differences in the presence of resistant E. coli between matrices/seasons/ locations.
I refer on the research below to point the authors on their work in future:
Monitoring of Water Quality, Antibiotic Residues and Antibiotic-Resistant Escherichia coli in the Kshipra River in India over a 3-Year Period; Nada Hanna, Manju Purohit, Vishal Diwan, Salesh P. Chandran, Emilia Riggi, Vivek Parashar, Ashok J. Tamhankar and Cecilia Stålsby Lundborg; International Journal of Environmental Research and Public Health; Published: 22 October 2020
Answer: Thank you for your comments and suggestions for improvement. With this help, the manuscript could be significantly improved and errors were discovered that we had unfortunately overlooked. In this study, we focused primarily on the comparison of water and biofilm. In particular, whether or not there is a difference in the population in terms of antibiotic resistance. It is of course clear that external factors have a different influence on the two populations and, as the reviewer noted, additional findings would make the picture clearer. Thanks also for the reference to this interesting publication with a similar question. We hope that we have now sufficiently cleared up all the uncertainties and errors.
R2: Line 18 Please, explain why you didn't mention the biofilm in the title?
Answer: We have changed the title.
R2: Line 31 Please, explain do you consider sediment and biofilm to be the same matrix?
Answer: We chose sediment as sample material because it was present at all sample sites and tried to extract the biofilm present in the sediment. So the biofilm grew on the sediment. That is why we have listed both terms together, even if they are not the same. That's why we refer to water and sediment when it comes to the place of origin. Biofilm and water population as a description of the way of living.
R2: Line 38/39 Please, delete the repeated sentence from Line 34/35.
Answer: Sentence was deleted.
R2: Line 100 Please, explain do you consider that samples which are scraped from a stone as biofilm? Please, reconsider for another term "Periphyton".
Answer: Sorry this is a relic of an early stage of the manuscript. At one point we also tried to extract biofilm from stones. This was not very successful and we then decided not to include these few isolates in the study. Thus, all biofilm isolates in this study came from sediment.
R2: "Periphyton or fouling represents an integral and independent microecosystem in aquatic ecosystems, which carries biotic components such as algae, fungi, bacteria, protozoa, metazoa together with abiotic components such as substrate, extracellular polysaccharides and detritus."
Answer: In bacteriology, the more commonly used term "biofilm" is used, even if these two terms are not synonymous.
R2: Line 116 Please, explain abbreviation COL?
Answer: We have replaced the abbreviation with the full name in the manuscript: Chromo Cult Coliform Agar
R2: Line 122 Did you confirm E. coli after they grew on blood agar? Which test - MALDI TOF?
Answer: Thanks for the comment. the sentence “Species identification for isolates was performed through MALDI-TOF mass spectroscopy” has been moved to the paragraph to make the correct flow of the experiments clearer.
R2: Line 125-129 Please, explain at what depth did you grab the sediment? How the samples from the stones were collected and on what surface?
Answer: Information added: It was ensured that mainly the superficial part of the sediment was harvested and a maximum of 5 cm deep.
R2: According to which protocol or publication did you process samples of sediment and stones? Please provide the reference.
Answer: We used a lab internal protocol which is described under 2.3. E. coli isolation from sediment samples
R2: Line 140 Please, explain how did you prepare the E. coli suspension for Muller Hinton agar? What turbidity did you use?
Answer: Information was added in line 151-154: two mL of sterile sodium chloride solution (0.9 %) were inoculated with one to three single colonies per isolate of fresh overnight cultures. The turbidity of these bacterial suspensions was adjusted to a turbidity of 0.5 ± 0.05 McFarland standard (DensiCheck, Biomérieux, Austria)
R2: Line 165-169 Please, explain which PCR detection and gene identification you performed?
Answer: Information was added in Line 185-192: “In brief, DNA was extracted by boiling of one colony suspended in 50 μl double-deionized water (95°C for 10 min.) After centrifugation for 1 min at 13000 rpm (Centrifuge 5415 R, Eppendorf) supernatant was used for PCR—reaction. Standard PCR protocols and conditions were modified in the following way: initial denaturation at 94°C for 5 min; 35 cycles at 95°C for 30 sec, 52°C for 45 sec, and 72°C for 60 sec; and final incubation for 10 min at 72°C using Taq DNA polymerase and dNTPs from QIAGEN (Hilden, Germany). Sequencing was performed with Eurofins overnight sequencing service (Eurofins, Germany).”
The Name of the genes can be also found in section 2.6 (blaCTX-M-1-group, blaCTX-M-2 -group, blaCTX-M-9-group, blaSHV, blaTEM and blaKPC)
R2: Line 172 Please, write the full name of the abbreviation "OD620".
Answer: The optical density at 620nm was added.
R2: Line 185 Please, reconsider whether Table 1 should be referred here, given that you are listing data from Table 2.
Answer: Mistake has been corrected.
R2: Table 2 Please, explain why in this table there is no separate data for E. coli strains isolated from biofilm as you stated.
Answer: See comments above. (We choose sediment as sample material because it was present at all sample sites and tried to extract the biofilm present in the sediment. So the biofilm grew on the sediment. That is why we have listed both terms together, even if they are not the same. That's why we refer to water and sediment when it comes to the place of origin. Biofilm and water population as a description of the way of living)
R2: Line 190-204 The results described in this paragraphs do not correspond to the data in Figure 1 (Panels A and B). Consider making separate figure for the described results.
Answer: The error has been corrected and a new Figure 1 has been added.
R2: Line 216-235 Please, reconsider to create a table with the results from the text in order to highlight statistically significant differences which you should then explain in the discussion.
Answer: A table with all proportions of resistance and multi-resistance in E. coli isolates from Mur and Drava water and sediment as well as the results (p-values) of the statistical tests was added as a new table 3.
Reviewer 3 Report
Comments and Suggestions for Authors
In the study, E. coli bacteria from two rivers were isolated from the flowing water and the sediment and characterized. Species identification was carried out by MALDI analysis. Antibiotic resistance genes were analyzed at the genotypic level and antibiotic resistance at the phenotypic level. Furthermore, the isolates were characterized with regard to metabolic reactions. Regardless of the river, the sample and the sampling site upstream or downstream of a wastewater treatment plant, there was no major difference in the numerical presence and characteristics of E. colis. It is a nice study of a characterization of the occurrence and the different E. colis in different streams.
As there were no significant differences in E. colis in the different samples, some further information regarding the areas of influence of the rivers would be necessary and should be included in the discussion:
How far away are the larger cities from the sampling sites? Are the two cities Graz and Villach of similar size in terms of population?
The difference between sediment and flowing water proved to be small. Is it possible that the result is due to the flow velocity? Is it low or high? Do both rivers have the same flow velocity? Do both rivers have a similar width?
These points should be discussed and debated in the discussion section.
Here some detail remarks:
- Line 13: Antibiotic-resistant, facultative….
- Lines 31f and 35f: content nearly identical, please shorten
- Line 62: other substances e.g. heavy metals and toxins ….. There are many other substances which are toxic, not only heavy metals.
- Line 97: please add the direction of the water flow during sampling: upstream or downstream.
- Line 102ff: the samples were taken in November/December and in April. Do you have data regarding weather and temperature at these time points? This may have an impact on the presence of E. coli in water.
- Line 119: please explain why you have incubated at 42°C, because this is not the optimal temperature for E. coli growth. You reduce the temperature to 37°C in the next step – please explain.
- Line 132: do you mean MALDI-TOF mass spectometry?
- Line 140: please add the company for the Müller-Hinton agar plates
- Line 154: I think a word is missing in this sentence
- Line 174: delete the comma
- Line 191: The E. coli isolates from Mur River….
- Line 192: delete … of isolates….
- Line 351: which “previous research” do you mean? Please add a reference.
- Line 353: …. river water. In the current…..
Moderate editing of English language is required
Author Response
R3: In the study, E. coli bacteria from two rivers were isolated from the flowing water and the sediment and characterized. Species identification was carried out by MALDI analysis. Antibiotic resistance genes were analyzed at the genotypic level and antibiotic resistance at the phenotypic level. Furthermore, the isolates were characterized with regard to metabolic reactions. Regardless of the river, the sample and the sampling site upstream or downstream of a wastewater treatment plant, there was no major difference in the numerical presence and characteristics of E. colis. It is a nice study of a characterization of the occurrence and the different E. colis in different streams.
As there were no significant differences in E. colis in the different samples, some further information regarding the areas of influence of the rivers would be necessary and should be included in the discussion:
How far away are the larger cities from the sampling sites? Are the two cities Graz and Villach of similar size in terms of population?
The difference between sediment and flowing water proved to be small. Is it possible that the result is due to the flow velocity? Is it low or high? Do both rivers have the same flow velocity? Do both rivers have a similar width?
These points should be discussed and debated in the discussion section.
Answer: Thank you for the provided review. With its help we were able to significantly improve the manuscript. The new version should answer all questions as best as possible and be (largely) free of mistakes. Thanks also for the hint about the river features. We have now integrated the information about the rivers into the manuscript. MaM (Line 93-121) and Discussion (line 405-413)
R3: Here some detail remarks:
R3: Line 13: Antibiotic-resistant, facultative….
Answer: Was changed.
R3: Lines 31f and 35f: content nearly identical, please shorten
Answer: Was done. One Sentence was deleted.
R3: Line 62: other substances e.g. heavy metals and toxins ….. There are many other substances which are toxic, not only heavy metals.
Answer: “pesticides” have been added to the list (and also added to the citations)
R3: Line 97: please add the direction of the water flow during sampling: upstream or downstream.
Answer: The following sentence has been added at the end of the paragraph: All samples were taken from the left bank of the river in the direction of flow.
R3: Line 102ff: the samples were taken in November/December and in April. Do you have data regarding weather and temperature at these time points? This may have an impact on the presence of E. coli in water.
Answer: The paragraph was expanded to include the physico-chemical water parameters at the respective sampling site. (Line 93-121)
R3: Line 119: please explain why you have incubated at 42°C, because this is not the optimal temperature for E. coli growth. You reduce the temperature to 37°C in the next step – please explain.
Answer: This temperature was chosen to reduce the accompanying flora during selection. E. coli usually survives incubation at 42°C, whereas bacteria in water do not (but would still grow at 37°C). For this reason, 42°C incubation is more frequently used for isolations of E. coli from water.
R3: Line 132: do you mean MALDI-TOF mass spectometry?
Answer: Yes. Line was relocated to be more precise.
R3: Line 140: please add the company for the Müller-Hinton agar plates
Answer: Manufacturer - bioMérieux, Austria - has been added.
R3: Line 154: I think a word is missing in this sentence
Answer: Sentence was rewritten: To determine the minimum inhibitory concentrations (MIC) for imipenem and meropenem, Etest® (bioMérieux Austria GmbH, Vienna, Austria) was used for all isolates that were resistant to at least one of the carbapenems tested.
R3: Line 174: delete the comma
Answer: Was deleted.
R3: Line 191: The E. coli isolates from Mur River…
Answer: Was added.
R3: Line 192: delete … of isolates….
Answer: Was deleted
R3: Line 351: which “previous research” do you mean? Please add a reference. Line 353: …. river water. In the current….
Answer: Citations were added.
Round 2
Reviewer 3 Report
Comments and Suggestions for Authors
All remarks and critical comments have been sufficiently completed. My proposal is to accept this manuscript for publication.